# Role of HIKESHI on Hyperthermia for Castration-Resistant Prostate Cancer and Application of a Novel Magnetic Nanoparticle with Carbon Nanohorn for Magnetic Hyperthermia

**DOI:** 10.3390/pharmaceutics15020626

**Published:** 2023-02-13

**Authors:** Takashi Nagai, Noriyasu Kawai, Masakazu Gonda, Keitaro Iida, Toshiki Etani, Daichi Kobayashi, Taku Naiki, Aya Naiki-Ito, Ryosuke Ando, Sataro Yamaguchi, Yuto Sugahara, Sakyo Ueno, Kaname Tsutsumiuchi, Toyoko Imae, Takahiro Yasui

**Affiliations:** 1Department of Nephron-Urology, Graduate School of Medical Sciences, Nagoya City University, Nagoya 467-8601, Aichi, Japan; 2Experimental Pathology and Tumor Biology, Nagoya City University, Nagoya 467-8601, Aichi, Japan; 3Center of Applied Superconductivity and Energy Research (CASER), Chubu University, 1200 Matsumoto, Kasugai 487-8501, Aichi, Japan; 4College of Bioscience and Biotechnology, Chubu University, 1200 Matsumoto, Kasugai 487-8501, Aichi, Japan; 5Graduate Institute of Applied Science and Technology, National Taiwan University of Science and Technology, Taipei 10607, Taiwan

**Keywords:** carbon nanohorn, hyperthermia, prostate cancer, HIKESHI

## Abstract

The prognosis of castration-resistant prostate cancer (CRPC) is technically scarce; therefore, a novel treatment for CRPC remains warranted. To this end, hyperthermia (HT) was investigated as an alternative therapy. In this study, the analysis focused on the association between CRPC and heat shock protein nuclear import factor “hikeshi (HIKESHI)”, a factor of heat tolerance. Silencing the HIKESHI expression of 22Rv1 cells (human CRPC cell line) treated with siRNAs inhibited the translocation of heat shock protein 70 from the cytoplasm to the nucleus under heat shock and enhanced the effect of hyperthermia. Moreover, a novel magnetic nanoparticle was developed via binding carbon nanohorn (CNH) and iron oxide nanoparticle (IONP) with 3-aminopropylsilyl (APS). Tumor-bearing model mice implanted with 22 Rv1 cells were examined to determine the effect of magnetic HT (mHT). We locally injected CNH-APS-IONP into the tumor, which was set under an alternative magnetic field and showed that tumor growth in the treatment group was significantly suppressed compared with other groups. This study suggests that HIKESHI silencing enhances the sensitivity of 22Rv1 cells to HT, and CNH-APTES-IONP deserves consideration for mHT.

## 1. Introduction

Prostate cancer (PC) is one of the most common cancers in men. Globally, approximately 1.4 million new cases are diagnosed every year [1]. The clinical strategy of localized PC is decided based on the risk stratification which consists of serum prostate-specific antigen (PSA) level, pathological characteristics, and clinical T-stage [2]. A 5-year survival rate of localized PC is approximately 99%, and has good prognosis [3]. However, the prognosis of locally advanced PC and PC with metastasis remains poor. Androgen deprivation therapy (ADT) is one of the main treatments of PC; however, most patients develop ADT resistance and castration-resistant prostate cancer (CRPC) within several years [4,5]. Available treatments for CRPC consist of chemotherapy or new-generation anti-androgen drugs; however, patients often develop drug resistance and poor prognosis [5]. Therefore, novel treatments for CRPC remain warranted.

Hyperthermia (HT) is a cancer treatment strategy based on the fact that the viability of cancer cells decreases depending on the temperature and time of heat shock [6]. Evidence for radiotherapy with HT against localized PC has been established [7,8], while for chemotherapy with HT, it is lacking. Moreover, HT, as a single-modality therapy for PC, is not a common treatment strategy due to several factors such as heat tolerance [9]. Heat shock proteins (HSPs) are heavily involved, as “molecular chaperones”, in the mechanisms underlying heat tolerance [10]. HSPs maintain protein homeostasis by repairing damaged proteins following cellular stresses such as heat shock, infection, or cancer [11]. In addition, HIKESHI, a protein encoded by human C11orf73, has also been reported as a transporter that supports the movement of HSP 70 to the nucleus under heat stress [12], and is a mediator of molecular chaperones that collaborates with various HSPs to protect cells against heat stress [13]. Overcoming heat tolerance is important to enhance HT effectiveness and outcome. However, data on heat tolerance of prostate cancer are lacking; therefore, this study examines the association between heat tolerance and HIKESHI expression in prostate cancer. 

The physical location of the prostate is another hindrance for HT as a treatment strategy against PC. Various methods, including ultrasound, radiofrequency, and lasers, have limitations when considering depth or focus [14], which can be overcome using magnetic HT (mHT), an antitumor therapy in which magnetic nanoparticles (MNPs) are accumulated at the cancer site while an alternative magnetic field (AMF) is applied from the outside of the body to locally heat the tumor. mHT utilizes the characteristic of MNPs, which generates heat under AMF [15,16]. The amount of heat depends on many factors such as the concentration of Fe in MNPs, physicochemical properties of MNPs, dispersion media, MNPs’ agglomeration state, and the frequency and intensity of AMF [16]. However, caution must be exercised when using AMF since water in the human body is a conductor; therefore, AMF can induce eddy currents that could damage otherwise healthy tissues [17]. The development of novel MNPs can enhance the efficiency of mHT. Previously, we developed MNPs to improve an effective mHT for PC. Magnetic cationic liposomes (MCLs) have been examined for their therapeutic effects against PC in vivo [18,19,20]. To develop a composite treatment using several therapies simultaneously, carbon nanohorn (CNH) is used, which is a horn-shaped sheath aggregate of graphene sheets discovered in 1999 [21]. CNH has several applications in cancer therapy including being an anticancer agent, drug delivery system, and gene therapy [22]. A novel MNP was developed using CNH containing a chemotherapy-drug, and the emission of the drug was promoted by mHT [23]. Another novel MNP conjugate was developed to obtain good dispersibility and exothermic properties in the present work. The effectiveness of mHT with MNPs consisting of CNH on prostate cancer remains unexplored. Achieving mHT with MNPs consisting of CNH can contribute to developing multimodal therapy including HT and other treatments. As an initial starting point to develop multimodal therapy for PC, the effectiveness of mHT against PC, using novel MNPs consisting of CNH, is examined. 

This study is focused on the mechanism of HT, related to HIKESHI, against PC in vitro and tissue microarray for human PC to examine HIKESHI and the Gleason pattern and evaluates the efficacy of mHT using novel MNPs consisting of CNH in vivo.

## 2. Materials and Methods

### 2.1. Cell Line

Human CRPC cell lines were used in this study; 22Rv1 (human PC cell line) cells, obtained from American Type Culture Collection (Rockville, MD, USA), were cultured in RPMI-1640 medium (Gibco Laboratories, Grand Island, NY, USA) supplemented with 5% fetal bovine serum and 1% penicillin/streptomycin and incubated at 37 °C in a humidified atmosphere of 5% CO_2_ and 95% air. 

### 2.2. Suppressing HIKESHI Using Small Interfering RNA (siRNA) Transfection

Two types of siRNAs for HIKESHI (Life Technologies, Carlsbad, CA, USA) and a negative control siRNA (Life Technologies) were obtained; 22Rv1 cells (5 × 10^5^) were seeded into 6-well plates and incubated at 37 °C overnight. After incubation, the cells were incubated in Opti-MEM^®^ I Reduced Serum Medium (Life Technologies) containing 20 nM siRNA and Lipofectamine™ RNAiMAX (Life Technologies) at 37 °C and maintained at 37 °C for 1 day.

### 2.3. Reverse Transcription-Quantitative PCR (RT-qPCR) Assay

Total RNA was extracted from cells using an RN easy Micro Kit (Qiagen, Hilden, Germany). Complementary DNA (cDNA) was produced from the reverse transcription of total RNA using a High-Capacity cDNA Reverse Transcription Kit (Life Technologies). RT-qPCR was performed on a 7500 Fast Real-time PCR system (Life Technologies) using a Taqman Fast Advanced Master Mix (Life Technologies). Glyceraldehyde 3-phosphate dehydrogenase (GAPDH) (Cell Signaling Technology, Danvers, MA, USA) was used for normalization.

### 2.4. Western Blot Analysis

Proteins were extracted by washing cells with phosphate-buffered saline and lysed with sodium dodecyl sulphate buffer. Equal amounts of each total protein (20 μg) lysate sample were dissolved in 12.5% polyacrylamide gels (ATTO corporation, Tokyo, Japan) and transferred onto Immobilon-P membranes (Merck, Darmstadt, Germany). The membranes were subsequently probed with horseradish peroxidase-conjugated secondary antibodies and visualized using an enhanced chemiluminescence kit (Global life science technologies, Tokyo, Japan). Antibodies against HSP70 (Stress Marq Biosciences Inc., Victoria, BC, Canada) and HIKESHI (Proteintech, Rosemont, IL, USA) were used to assess protein expression levels. GAPDH (Cell Signaling Technology, Massachusetts, USA) and anti-Histone H3 (Abcam, Cambridge, UK) were used as a loading control.

### 2.5. Cell Viability Assay

A cell counting kit (Dojindo Co, Kumamoto, Japan), a WST-8 based assay, was used to assess cell viability. Cells were incubated with the WST-8 solution at 37 °C. After 2 h, the concentration of formazan dye was determined via absorbance at 450 nm. 

### 2.6. Tissue Microarray of Human Prostatectomy Specimens 

Prostatectomy specimens, obtained from the Aichi Cancer Center Hospital between 2009 and 2010, were examined. All specimens were analyzed after written informed consent was obtained from the patients according to an Institutional Review Board approved protocol with approval number 1168. All cases were evaluated by a panel of experienced pathologists and rescored according to the Gleason pattern. Tissue arrays were prepared from formalin-fixed, paraffin-embedded tissue specimens of 46 prostate cancer patients.

### 2.7. Immunohistochemical Analysis

The mouse tissues were also incubated with 1:100 diluted anti-HSP70 (StressMarq Biosciences Inc., Victoria, BC, Canada), 1:1000 diluted anti-HIKESHI (Proteintech, Rosemont, IL, USA) antibody, or anti-Ki-67 (abcam, Cambridge, UK). 

Antibody binding was visualized using a conventional immunostaining method with a Leica bond max autoimmunostaining apparatus (Leica, Wetzlar, Germany). 

The labeling index of Ki-67 was analyzed by counting at least 1000 cells under a microscope at high magnification using BZ-9000 multifunctional microscopy (Keyence corporation, Osaka, Japan) and Image J, an image analysis software [24]. The relative intensity score of HIKESHI expression was evaluated in the normal prostate glands and carcinoma cores in each specimen. The human prostate microarray specimens were allocated into three groups; (1) normal glands and Gleason pattern 3 group, (2) Gleason pattern 4 group, and (3) Gleason pattern 5 group (Normal glands and Gleason pattern 3: n = 139, Gleason pattern 4: n = 99, Gleason pattern 5: n = 5). The raw cytoplasm intensity data for luminal cells in normal prostate glands and tumor cells in PC cores were evaluated according to the Gleason pattern. In each core, five glands of the intensity score were measured using BZ-9000 multifunctional microscopy and analysis software (Image J).

### 2.8. Preparation of CNH-APS-IONP

The synthetic method consists of three stages: oxidation of CNH, preparation of iron oxide nanoparticle (IONP), and conjugation with 3-aminopropyltriethoxysilane (APTES). 800 mg of CNH (as-grown, NEC Corporation, Tokyo, Japan) was added to 200 mL of concentrated nitric acid and stirred vigorously for 1 h under reflux condition using a stirring heating mantle. The resulting black dispersion was centrifuged (14,970× *g*, 10 min), and the precipitate was vortexed with water and centrifuged again. The dispersion of the precipitate in water was filtered through a spin filter Amicon^®^ Ultra-15 (molecular weight cut-off [MWCO] 100k Dalton; Merck, Darmstadt, Germany) and washed thoroughly with water until the pH of the filtrate reached 5–6. The resulting solution was lyophilized to obtain CNH_OX_ powder (Figure 1A). 

IONP was prepared by adding aqueous ion solution (300 mL, 2.0 mol/L, FeCl_2_:FeCl_3_ = 1:2) to sodium hydroxide solution (700 mL, 4.0 mol/L) with a vigorous mixing by a homogenizer for 2–3 min at room temperature. The reaction mixture was centrifuged at 7485× *g* for 1 min, and then the precipitate was collected, 150 mL of water was added and homogenized, and the precipitate was collected via centrifugation at 7485× *g* for 1 min. Next, after zirconia balls was added to the dispersion, the mixture was sonicated with an ultrasonic homogenizer UH-600 (SMT Co., Tokyo, Japan) (10 min × 4) and shaken in a 500 mL polypropylene (PPCO) bottle (Thermo Scientific, Waltham, MA, USA) using a multi-shaker MMS-3010 (EYELA, Tokyo, Japan) for a week. Then, the supernatant was filtered through a 0.22 µm filter (Iwaki bottle top filter, 500 mL, PES; AGC Techno Glass Co., Shizuoka, Japan) after centrifugation at 14,970× *g* for 10 min (Figure 1B). 

To prepare APS-IONP, the IONP dispersion (IONP content, 5.0 g) was lyophilized and sonicated for 15 min in 80 mL toluene-ethanol mixture (1:1 [*v*/*v*]). The particle dispersion was transferred to a PPCO bottle, and zirconia balls were added. Furthermore, 5.0 mL of 25% ammonia aqueous solution and 5.0 mL (21 mmol) of APTES were added and the mixture was shaken for 1 day at room temperature. Then, the magnetic particles were washed three times with ethanol and water (Figure 1C); 750 mg of APS-IONP was dissolved in 30 mL of water, which was degassed by nitrogen bubbling, and the mixture was dispersed by sonication for 2 min. To the CNH_OX_ dispersion, 70.0 mg (0.253 mmol) of 4-(4,6-dimethoxy-1,3,5-triazin-2-yl)-4-methyl-morpholinium chloride (DMT-MM) was added and shaken for 1 h, following which the APS-IONP dispersion was added. The reaction dispersion was shaken at room temperature throughout the day, a small amount of sodium hydroxide solution was added to adjust the pH of the reaction solution to 8–9, and it was centrifuged (14,970× *g* for 10 min). The resulting supernatant was removed using a spin filter Amicon^®^ Ultra-15 (MWCO 100 k) to collect the solid of CNH_OX_-APS-IONP (Figure 1D).

### 2.9. Morphological Features of CNH-APS-IONP

Energy dispersive X-ray (EDX) spectroscopy, transmission electron microscopy (TEM), and dynamic light scattering (DLS) were performed to assess morphological features of CNH-APS-IONP. Meanwhile, infrared (IR) absorption spectrometry was performed to characterize the chemical compounds using a KBr method with FT/IR-4700 (JASCO, Tokyo, Japan). The distribution of each particle and atom of CNH-APS-IONP was observed using JEM-2100F (JEOL, Tokyo, Japan), while the particle size of each compound was assessed via DLS using FPAR-700 (Otsuka Electronics, Osaka, Japan). 

### 2.10. Animal Models and Therapy Protocol

Six-week-old male BALB/c nude mice were purchased from Japan SLC (Hamamatsu, Japan). A mouse model for subcutaneous implantation of prostate cancer was used. To prepare tumor-bearing animals, cell suspensions containing approximately 1 × 10^7^ 22Rv1 cells in 100 μL of phosphate buffer (0.05 M sodium phosphate and 0.15 M NaCl, pH 7.4) mixed with Matrigel (Corning incorporated, Corning, NY, USA) were injected subcutaneously into the right flank of BALB/c nude mice under short-term anesthesia by intraperitoneal injection (i.p.) of sodium pentobarbital (50 mg/kg body weight). Tumor volume was measured using caliper and estimated as π × length × width × height/6. 

Three weeks (day 21) following subcutaneous implantation, tumor-bearing mice were allocated into three groups namely control (n = 6), non-treatment (n = 6), and treatment (n = 6). The mice in the control group were not injected with CNH-APS-IONP or set under AMF; the mice in the non-treatment group were injected locally via focal injection with 300 μL of CNH-APS-IONP (Fe concentration: 36.3 mg/mL) around the tumor site on day 21 but not set under AMF. The mice in the treatment group underwent HT as follows: Mice were anesthetized with subcutaneous injection of combined anesthetics with dexmedetomidine, hydrochloride, midazolam, and butorphanol tartrate mixed according to the protocol reported in the previous literature [25] and injected locally with 300 μL of CNH-APS-IONP (Fe concentration: 36.3 mg/mL) via focal injection around the tumor on day 21 (Figure 2A). Thereafter, HT from the heat generated by CNH-APS-IONP under an AMF was attempted using a coil-type AMF device on days 21 and 22 (Figure 2B,C). The temperature in the tumor was maintained at 42–46 °C, and the heating was performed for 15 min. Following treatment, a subcutaneous injection of atipamezole was completed as the antagonist drug for anesthesia. The tumor size in both groups was measured over time to verify the effect of HT on days 25 and 28. Animal experiments were performed according to the principles laid down in the Guide for the Care and Use of Laboratory Animals prepared under the direction of the Office of the Prime Minister of Japan. All experimental procedures were approved by the laboratory animal facility, Graduate School of Medical Sciences, Nagoya City University (IDO 19-041).

### 2.11. AMF Generator

The mouse-sized AMF generator with five small coils and power oscillators, which was developed for HT, is lab-made (Figure 2C) [26]. This generator is operated by the direct current power supply, and the electric current of the coil depends on its voltage and current. The frequency was 100 kHz and the magnetic field amplitude at the center of the coil was >35 kA/m (44 mT). 

### 2.12. Temperature Measurements

The rectal temperature of the mice was measured using a fiber thermometer (Anritsu Co., Tokyo, Japan) which does not contain metal. Thermography (FLIR C5, FLIR systems, Wilsonville, OR, USA) was used simultaneously to measure the temperature of the tumor (Figure 2D). Rectal and tumor temperatures were recorded by the minute until the temperature of the tumor reached 42 °C and then every 5 min.

### 2.13. Statistical Analysis

Data are presented as the mean ± standard error. Statistical analysis was performed to assess the association between variables using Welch’s *t*-test or Kruskal-Wallis test using EZR [27]. Correlation for multiple comparison was made using Steel–Dwass post hoc test. A value of *p* < 0.05 was considered statistically significant. 

## 3. Results

### 3.1. Functional Analysis of HIKESHI Knockdown in 22Rv1 Cells

RT-qPCR and western blot analysis was performed to assess the ability of siRNAs for HIKESHI. RT-qPCR showed that the mRNA expression level of HIKESHI was inhibited by siRNAs compared with negative control cells (Figure 3A). Western blotting showed that HIKESHI was suppressed in siRNA for HIKESHI groups as compared with the negative control cells at 37 °C 24 h after heat shock and 48 h after heat shock (Figure 3B). Increased HIKESHI expression lasted for 48 h in the negative control cells. The localization of HSP70 and HIKESHI was also assessed under heat shock (Figure 3C). The HSP 70 protein levels in the nucleus of each band were quantified; heat shock treatment induced HSP 70 translocation from the cytoplasm to the nucleus. Moreover, suppressing HIKESHI resulted in inhibiting translocation of HSP 70 from the cytoplasm to the nucleus (Figure 3D).

### 3.2. Heat Shock Treatment on Cell Viability under HIKESHI Knockdown in 22Rv1 Cells

The cells were incubated for 3 h at 43 °C on days 0 and 1 as heat shock treatment after transfecting siRNAs. After heat shock treatment, cells were incubated at 37 °C. 

The viability of the cells was tested after 24 h. The therapeutic effects of heat shock were assessed using a cell viability test with the WST-8 assay (Figure 4). On day 0, there was no significant difference among the three groups. However, the cell growth was more suppressed in the HIKESHI siRNA groups than in the negative control group on days 1 and 2 (*p* < 0.001 and *p* < 0.001, respectively). 

### 3.3. Human Prostate Tissue Microarray of HIKESHI

HIKESHI expression in the Gleason pattern 4 group and Gleason pattern 5 group was significantly higher than normal glands and the Gleason pattern 3 group. Moreover, the intensity of the Gleason pattern 5 group was significantly higher than that of Gleason pattern 4 group (Figure 5).

### 3.4. Morphological Features of CNH-APS-IONP

In this study, a novel CNH-IONP conjugate, CNH-APS-IONP, was developed. It was prepared with APTES to make the chemical bond between CNH and IONP. The IR spectra of APS-IONP and CNH-APS-IONP showed bands around 560 cm^−1^ (assigned to Fe-O stretching), 1000 cm^−1^ (Si-O stretching), 1200 cm^−1^ (C-C stretching), 1600 cm^−1^ (C=O [-{CONH-}] stretching), 1700 cm^−1^ (C=O [COO^−^] stretching), and 3200–3500 cm^−1^ (O-H [COOH] and N-H [-CONH-] stretching) (Figure 6A). 

The TEM image of CNH-APS-IONP showed one CNH particle bonded the multiple IONPs (Figure 6D). In addition, EDX analysis revealed that the presence of IONP (approximately 5–10 nm) was confirmed by the overlap of the green dots (mapping of Fe, Figure 6E) and red dots (mapping of O, Figure 6F) with the black circular image in the original. Moreover, the yellow dots (mapping of C, Figure 6G) were found along the CNH-APS-IONP in the original image. Therefore, the contour seen at front and background of IONPs was considered CNH. The particle sizes of CNHox, IONP, and CNH-APS-IONP measured via DLS are listed in Table 1. 

### 3.5. Magnetic Hyperthermia Using CNH-APS-IONP Suppressed Tumor Growth In Vivo

The body weight of each group is listed in Table 2. There were no significant differences among the three groups. 

The temperature course of the subcutaneously implanted model mice during the thermal treatment is shown in Figure 7A,B. The tumor temperature in all mice rose above 42 °C in approximately 5 min and remained above 42–46 °C for 15 min. The rectal temperature did not rise in any of the mice, and the tumor temperature of each mouse remained approximately 20 °C higher than that of the rectum under AMF for 2 days. The voltage of AMF was maintained between 45–60 V, and the electric current was maintained between 5.4–7.1 A (magnetic field, 18–25 kA/m (23–31 mT)). 

The time course of tumor size in the control group (n = 6), non-treatment group (n = 6), and treatment group (n = 6) is shown in Figure 7C. The tumor size of the treatment group was the lowest among the three groups. The growth ratio of the treatment group was significantly lower than that of the other two groups (Figure 7D). 

Hematoxylin and eosin (H&E) staining showed that the sites close to CNH-APS-IONP degenerated (Figure 8). Pyknosis and nuclear rupture were observed at the degenerated sites.

The Ki-67 labeling index of the treatment group was significantly decreased by mHT compared with the control and the non-treatment groups (Figure 9). 

## 4. Discussion

The association of HSP70 and HIKESHI for CRPC was analyzed in vitro, and the association between the expression of HIKESHI and the degree of malignancy in PC was examined. In addition, the effect of mHT with MNP using CNH for a CRPC model was verified. 

The results showed that the therapeutic effects of HT were enhanced by silencing HIKESHI. Upon inhibition of HSP 70 movement to the nucleus from the cytoplasm, cellular stresses such as heat, UV light, and oxidative stress inactivate transportation efficiency of proteins with a conventional nuclear localization system and nuclear-cytoplasm transportation via importin α and β families [28]. HIKESHI is a novel transporter that did not belong to the importin β families that delivered HSP70 from the cytoplasm to the nucleus under heat shock [12]. HIKESHI expression, first demonstrated in HeLa cells under heat shock [12], is key to heat tolerance in a gastric cancer cell line and human tongue squamous carcinoma cell line [29,30]. Promotion of HIKESHI expression was suggested to be via HSF1-independent transcriptional mechanisms [30]. Therefore, preventing HIKESHI expression enhances sensitivity to HT. Previous studies on HSPs inhibitors were effective in splicing a variant of the androgen receptor. HSP 90 and HSP 70 inhibitors were reported to suppress mRNA splicing of androgen receptor variant 7 in PC cells [31,32]. Suppressing HIKESHI expression can improve the therapeutic effects of HT. Further studies are warranted to explore the specific role of HIKESHI expression in PC. The present study investigated the association between HIKESHI expression and Gleason pattern via tissue microarray. Gleason pattern is a well-known marker for the malignancy of PC. Our results showed that HIKESHI expression correlated with the Gleason pattern, suggesting that PC with higher malignancy would serve higher thermos tolerance. In a case report of a patient with small cell PC and a residual lesion, HT ineffectively showed the overexpression of HIKESHI. HIKESHI c11orf73 was also reported as one of the gene-based biomarkers that overexpresses in the late stage of clear cell renal cell cancer [33]. In gastric cancer clinical specimens, lymphatic invasion was related to HIKESHI expression [29]. However, the expression of HIKESHI was not associated with cancer progression and prognosis in the study [29]. The expression of HIKESHI is a potential marker for the malignancy of PC. However, further studies are necessary to confirm the thermotolerance under HT.

Our results also demonstrated the effect of mHT for the CRPC mouse model using CNH-APS-IONP. The original AMF generator used for mice was lab-made. CNH-APS-IONP, a lab-made novel MNP, demonstrated high thermal stability in this study and was considered for HT. Moreover, tumor growth was suppressed by mHT using CNH-APS-IONP. To the best of our knowledge, this is the first study to treat CRPC in vivo using MNP consisting of CNH. The results of this study, where MCL for mHT was used, are comparable with those of previous studies [18,19,20]. The use of nanotechnology to diagnose or treat PC has attracted attention [34]. Synergy with nanotechnology and multimodal therapy including radiotherapy, chemotherapy, and gene therapy are expected [34]. One of the advantages of coupling CNH is the potential use as a drug carrier; hence, it has a significant development potential compared with IONP alone, which can only provide hyperthermia for PC. The utility of CNH for PC treatment has been assessed. In vitro, siRNA with CNH could knockdown specific genes and enhance the therapeutic effect of chemotherapy [35]. In addition, prostate-specific membrane antibody with CNH achieved specific effects against PC [36]. CNH-APS-IONP itself provides only hyperthermic effects; however, its therapeutic effect can be enhanced when coupled with CNH. Moreover, CNH-APS-IONP may contribute to other carcinomas if specific drugs or antibodies are bound to CNH [23].

This study has some limitations. First, examination of the CNH-only group and IONP-only group was not conducted. This study examined the effectiveness of mHT via CNH-based MNP, but the impact of coupling CNH on mHT remains to be examined, including whether CNH synergistically enhances the effects of mHT. Second, CNH-APS-IONP was injected via focal injection, which is effective for unifocal disease not multifocal metastasized disease as is the case for CRPC. 

Future work should include preparing samples with high iron concentration in MNPs and AMF conditions to obtain better therapeutic effects. Research from Nagoya City University has reported that subcutaneous PC completely regressed after multiple cycles of HT [20]; multiple cycles with appropriate treatment intervals should be further investigated. 

## 5. Conclusions

The association between HIKESHI and PC in vitro was analyzed. Suppressing HIKESHI expression may be a key to overcoming heat tolerance in HT. The clinical specimens of PC were analyzed; the results suggested an association between HIKESHI expression and malignancy and illustrated the importance of HIKESHI expression on HT in PC. Moreover, we developed a novel MNP, CNH-APS-IONP, and demonstrated the therapeutic effects of mHT on CRPC model mice in vivo. The MNP using CNH has extensibility owing to its function as a drug delivery system. Focusing on HIKESHI and MNP using CNH is the key to HT PC.

## Figures and Tables

**Figure 1 pharmaceutics-15-00626-f001:**
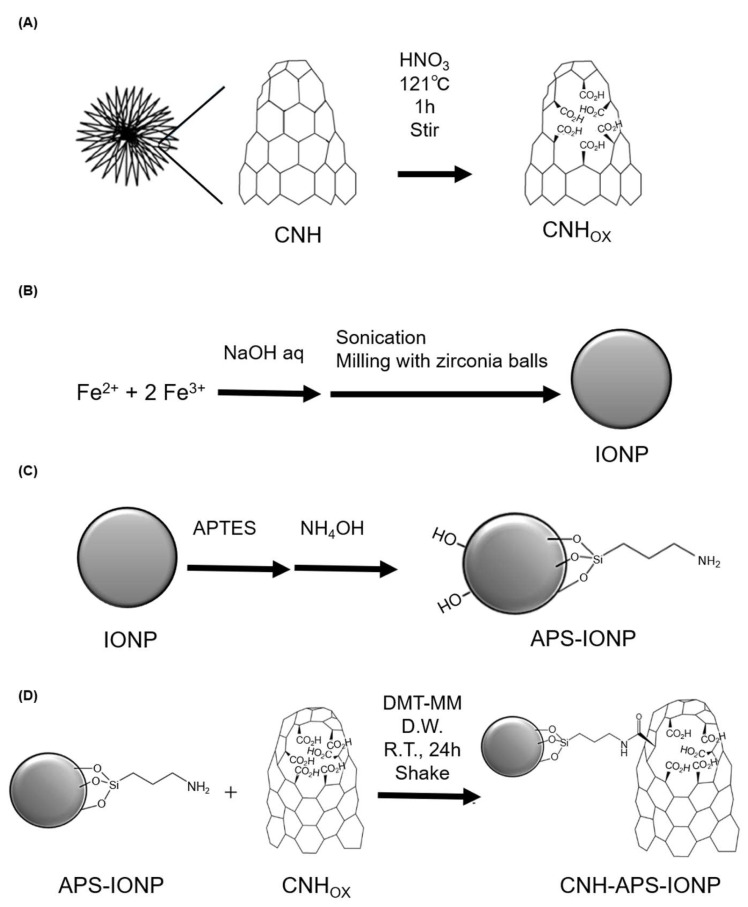
Chemical compounding process of CNH-APS-IONP. (**A**) CNH was added to nitric acid and stirred for 1 h at a dispersion temperature of 121 °C. (**B**) IONP was produced from Fe and zirconia balls. (**C**) APS-IONP was produced from APTES and IONP. (**D**) CNH-APS-IONP was produced from CNHox and APS-IONP. DMT-MM: 4-(4,6-dimethoxy-1,3,5-triazin-2-yl)-4-methyl-morpholinium chloride.

**Figure 2 pharmaceutics-15-00626-f002:**
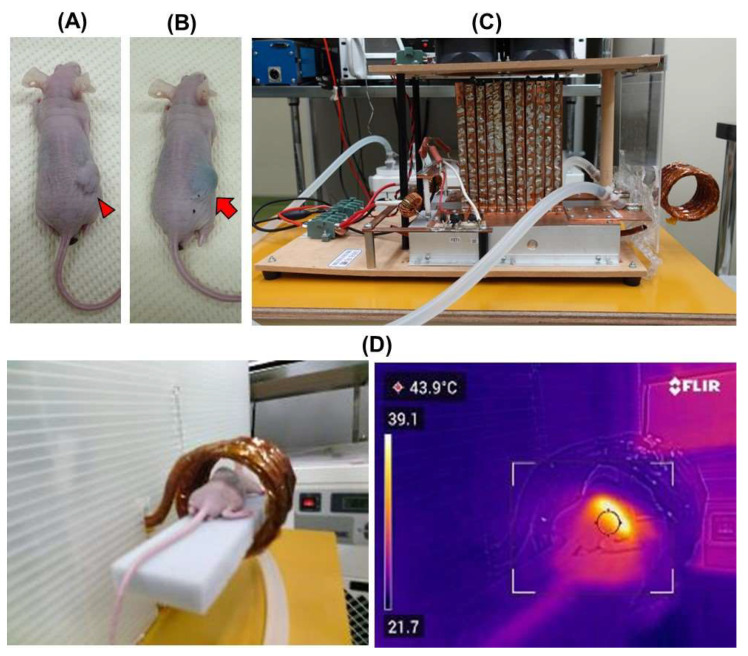
Representative images of therapy protocol. (**A**) The tumor on day 21, before CNH-APS-IONP injection (arrowhead). (**B**) The tumor on day 21, after subcutaneous CNH-APS-IONP injection (arrow). (**C**) AMF-generating device. AMF generates in the coils. (**D**) The image of thermography shows that heat generates around the tumor of a mouse placed in the coils under AMF.

**Figure 3 pharmaceutics-15-00626-f003:**
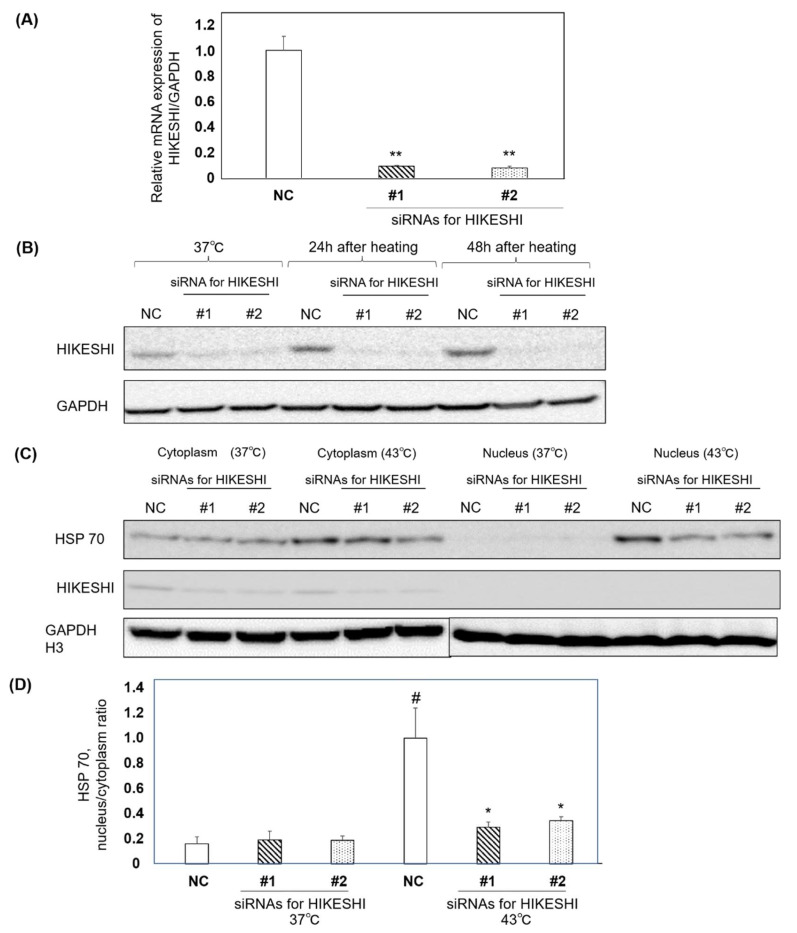
(**A**) Quantitative reverse transcriptase PCR of 22Rv1 cells treated with siRNA as NC and siRNAs for HIKESHI (n = 3 in each group). The expression of mRNA in siRNAs for HIKESHI was suppressed compared with NC. Data are presented as mean ± standard error of the mean (SE); ** *p* < 0.01 (Welch’s *t*-test). (**B**) Western blot analysis of HIKESHI expression in 22Rv1 cells after siRNA treatments. Regulation of HIKESHI in 22Rv1 cells. Cells were harvested after knockdown of HIKESHI without heat shock, at 24 h after heat shock, and at 48 h after heat shock. The suppression of HIKESHI lasted for at least 48 h. (**C**) Western blot analysis of HIKESHI expression in 22Rv1 cells after siRNA treatments. The location of HSP 70 after heat shock was assessed. The cells were immediately harvested after heat shock and divided into the nucleus and cytoplasm. (**D**) The expression of HSP 70 of each band was quantified, and the ratio (nucleus/cytoplasm) was calculated (n = 3). Data are presented as the mean ± standard error of the mean (SE); # *p* < 0.0001 vs. NC at 37 °C (Welch’s *t*-test); * *p* < 0.05 vs. NC at 43 °C (Welch’s *t*-test).

**Figure 4 pharmaceutics-15-00626-f004:**
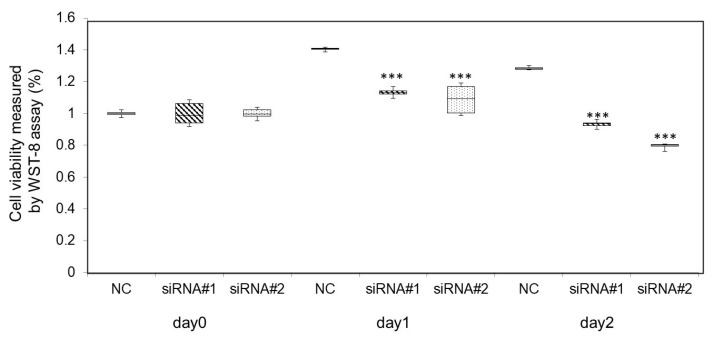
Effect of HIKESHI knockdown on viability in 22Rv1 cells using WST-8 assay. The cells treated with siRNAs for HIKESHI showed significantly higher sensitivity than those of the negative control. *** *p*-value < 0.001 vs. NC on each day (Welch’s *t*-test). NC, negative control.

**Figure 5 pharmaceutics-15-00626-f005:**
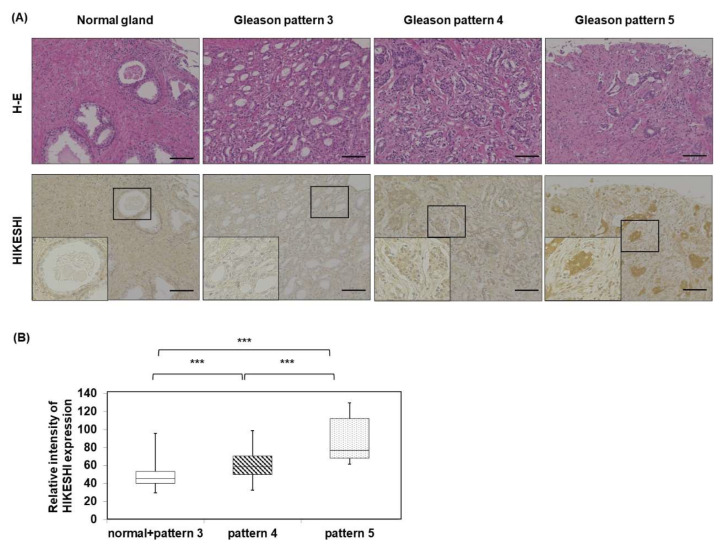
(**A**), Immunohistochemistry for HIKESHI in normal prostate normal glands, Gleason pattern 3 adenocarcinoma, Gleason pattern 4 adenocarcinoma, and Gleason pattern 5 adenocarcinoma. (**B)** The relative intensity of the normal glands and Gleason pattern 3 groups (n = 139), Gleason pattern 4 group (n = 99), and Gleason pattern 5 group (n = 5). *** *p* < 0.001. (Kruskal-Wallis test) Bar: 100 μm.

**Figure 6 pharmaceutics-15-00626-f006:**
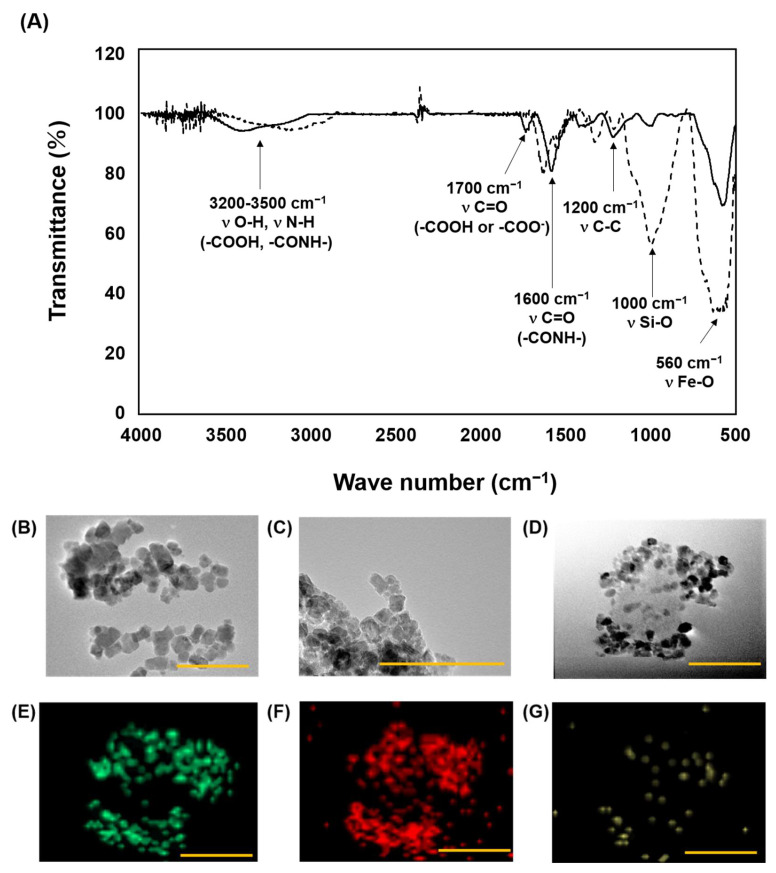
(**A**) Infrared absorption spectra of APS-IONP (dotted line) and CNH-APS-IONP (solid line). Transmission electron microscopy of (**B**) IONP, (**C**) APS-IONP, and (**D**) CNH-APS-IONP. Energy dispersive X-ray (EDX) spectroscopy of CNH-APS-IONP showing the mapping of (**E**) Fe, (**F**) O, and (**G**) C. Bars: 100 nm.

**Figure 7 pharmaceutics-15-00626-f007:**
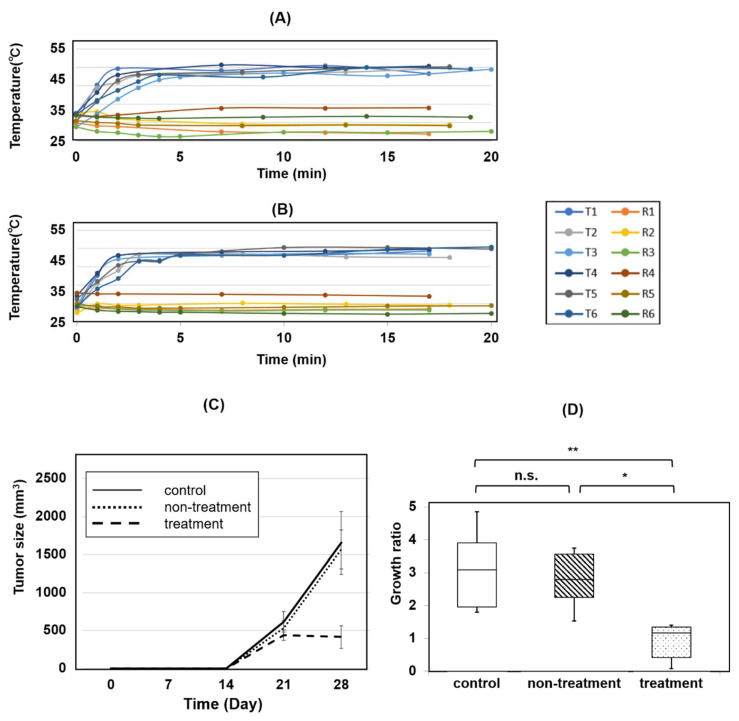
(**A**) The temperature of the treatment group (n = 6) on day 21. (**B**) The temperature of the treatment group (n = 6) on day 22. T, tumor temperature; R, rectal temperature; 1–6, number of each mouse. (**C**) Time course of tumor size in the control group (n = 6), non-treatment group (n = 6), and treatment group (n = 6). MNP was injected subcutaneously for the non-treatment group and the treatment group on day 21. AMF generation was conducted for the treatment group on day 21 and day 22. The average size of each group is shown as a line. Data are presented as mean ± standard error of the mean (SE). (**D**) Growth ratio of tumors among the three groups. The growth ratio was calculated based on the size of tumor on days 21 and 28. n.s., not significant; * *p* < 0.05; ** *p* < 0.01 (Kruskal-Wallis test).

**Figure 8 pharmaceutics-15-00626-f008:**
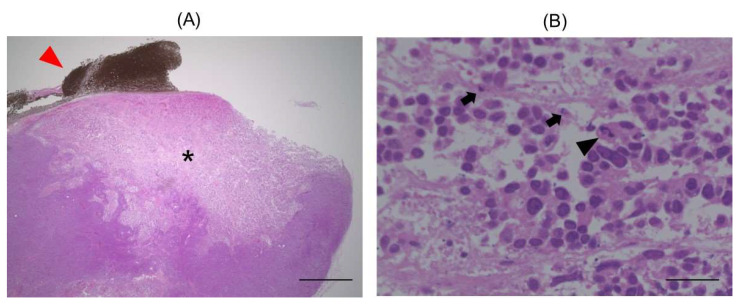
Representative hematoxylin-eosin staining image of the tumor that underwent hyperthermia. (**A**) A degenerated site (*) around CNH-APS-IONP (red arrowhead) was observed. (**B**) Pyknosis (arrow) and nuclear rupture (black arrowhead) are seen in the high-power fields. Bar: 1000 μm (**A**), 25 μm (**B**).

**Figure 9 pharmaceutics-15-00626-f009:**
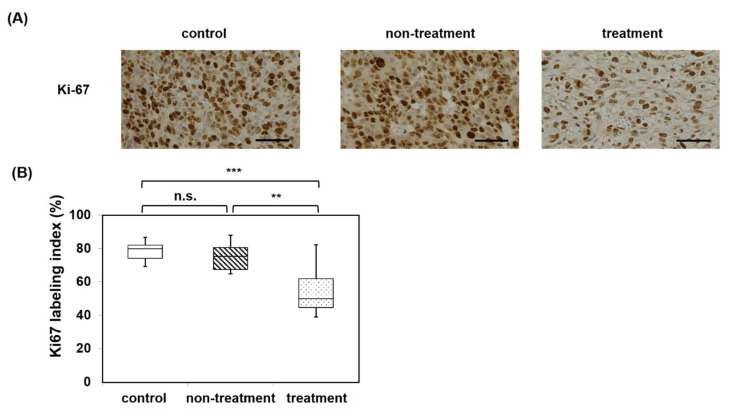
(**A**) Immunohistochemistry of Ki-67 of control, non-treatment, and treatment. (**B**) Ki-67 labeling index of the control group, the non-treatment group, and treatment group (n = 3, each group). The Ki-67 labeling index of the treatment group was the lowest among the three groups. No significant difference was observed between the control and non-treatment groups. n.s., not significant; ** *p* < 0.01; *** *p* < 0.001 (Kruskal-Wallis test) Bar: 50 μm.

**Table 1 pharmaceutics-15-00626-t001:** Particle size of each compound.

Sample	Particle Size, nm
CNHox	213 ± 24
IONP	79 ± 38
CNH-APS-IONP	238 ± 67

Data are presented as mean ± standard deviation (SD).

**Table 2 pharmaceutics-15-00626-t002:** Body weight in the different groups.

Time ^a^, Day	Body Weight, g			*p*-Value
	Control Group	Non-Treatment Group	Treatment Group	
0	20.9 ± 0.9	20.5 ± 0.7	20.9 ± 0.8	ns
7	21.5 ± 0.3	22.0 ± 0.5	21.4 ± 0.5	ns
14	21.8 ± 0.4	22.5 ± 0.4	22.2 ± 0.4	ns
21	22.8 ± 0.4	22.4 ± 0.7	22.5 ± 0.6	ns

^a^ Time after subcutaneous implantation of prostate cancer to BALB/c nude mice. Data are presented as mean ± standard error of the mean (SE). ns, not significant (Kruskal-Wallis test).

## Data Availability

All data supporting the findings of this study are presented in this manuscript.

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
