# Peer review of "Role of HIKESHI on Hyperthermia for Castration-Resistant Prostate Cancer and Application of a Novel Magnetic Nanoparticle with Carbon Nanohorn for Magnetic Hyperthermia"

_pharmaceutics, 2023, doi:10.3390/pharmaceutics15020626_

Round 1

Reviewer 1 Report

The submitted manuscript reports on the role of HIKESHI on hyperthermia for castration-resistant prostate cancer and application of a novel magnetic nanoparticle with carbon nanohorn for magnetic hyperthermia. This topic is of interest for readers of Pharmaceutics. However, I have some reservations about the experimental design and data presented. I therefore recommend the publication of this manuscript only if the authors can address the major issues noted below.

1. The introduction section needs to be improved. It needs to be clearly stated what are key research gaps, which are not achieved in the literature.

2. The experimental sections are quite problematic as the groups are not well designed and compared. Groups should include blank control, CNH, APS-IONP and CNH-APS-IONP. Currently it is extremely hard to compare the results as no proper controls are studied.

3. The physicochemical properties of the nanoparticles need to be characterized using a transmission electron microscope and dynamic light scattering for all the groups, not just an image of CNH-APS-IONP.

4. It is necessary to do cellular uptake study for the nanoparticles.

5. To understand the therapeutic effects, the authors should do tumor growth and body weight measurement for all different groups. These data are critical, but missing currently.

6. The ethical approval and the Committee approval number should be stated for studies on animals.

7. Extensive revision of the English and form to correct errors and typos in the manuscript. Also some words were in red color. Please make sure the format is consistent.

8. The information in some figures is difficult to read and needs to increase the size and resolution.

Author Response

Attached a file.

Reviewer 2 Report

Dear authors,

thank you for your interesting article about hyperthermia effect on prostate cancer model using nanohorn carbon coupled to iron oxides nanoparticles. I agree with the interest of the approach.

Nevertheless, I regret the fact your your work does not support some important goals you proposed to follow.

First, there is two parts on your text: one about iron-oxide aided magnetic hyperthermia, the other the pathophysiology of HIKESHI expression in prostate cancer. It would be more clear to split the both features in two articles.

About nanoparticles, you did not really justifify the use of coupling carbon nanohorn on iron oxide nanoparticles: you expected better results than iron oxide nanoparticles alone ? Did you look forward to use nanohorn as drug carrier for synergistic pharmacology - hyperthermia effects ? Then, iron-oxide alone reference groups (without and with hyperthermia) are necessary to compare the effect of nanohorn addition. It should be emphasized that direct intra-tumoral injection can be possible for unifocal disease but not for multifocal metastasized disease as usually seen in CRPC.

Some technical details woud be useful too for the understanding:

- for parametric statistical tests (Student's and variance analysis), did prove the normality of the data ? When did you used these tests (please precise in the text).

- how did you for intratumoral particles injection ? focal injection ? unifocal transdermic puncture with multiple deposits in the whole intra-tumoral volume ? multi-punctures ?

- what was the delay between particle injection and the Day 21 hyperthermia irradiation ?

- please correct references 5, 13 and 19 for homogeneity. 

Sincerly yours 

Author Response

Attasched a file.

Reviewer 3 Report

In the paper entitled “Role of HIKESHI on hyperthermia for castration-resistant pros-tate cancer and application of a novel magnetic nanoparticle with carbon nanohorn for magnetic hyperthermia” the authors reported the design of novel magnetic nanoparticles modified with carbon horns and their use as magnetothermal agents for the treatment of castration-resistant prostate cancer, in combination with the study of the effect of hikeshi silencing. I found the paper very well-detailed and organized, with good critical discussion and the adequate references to support the reached conclusions, being suitable for its publication in Pharmaceutics after polishing some minor details. Following I expose some comments and suggestions that could improve the paper and which I would like the authors to address before consider resubmission.

I suggest the design of a simple schematic figure/graphical abstract illustrating the work carried out in the paper.

“It utilizes the characteristic of magnetic nanoparticles (MNPs) contain-ing iron oxide nanoparticles (IONP)”. Nanoparticles containing nanoparticles? Rephrase sentence, please.

The state of the art of the problematic and the role of heat shock proteins are very well-explained in the intro, but I missed a more detailed explanation of magnetic hyperthermia phenomenon in the introduction. Mention please the magnetic behavior of the particles typically used for this purpose (superparamagnetic), the importance of NPs physicochemical properties (size, shape, composition, etc.) on their magnetothermal effect, the characteristics and limitations of the applied alternatig magnetic fields, etc. You can have a look in these recent papers published by GFCP group from USC: https://link.springer.com/chapter/10.1007/978-3-319-89878-0_7; ACS Appl. Mater. Interfaces 2020, 12, 8, 9017–9031 and Chem. Mater. 2020, 32, 6, 2220–2231

Report your centrifugation speeds in RCF to allow the reproducibility of your protocols.

Typo: Figure 3A, “1.0” value imoved in y-axis

Figure 5: I suggest to indicate the bond associated with each IR band in the graph and delete the table of figure 5B.

Increase resolution of Figure 6 and delete the definition of CNH, APS and IONP from the legend. They already defined. Moreover, I suggest combining figures 5 and 6 in only one panel to present nanoparticles characterization all together.

You reported the electric intensity and voltage in the coils used to generate the alternatig magnetic field, but did you measure or calculate the intensity of the magnetic field? How many miliTeslas?

Can you speculate a bit more in the discussion section the effect/importance of the modification of magnetic NPs with carbon horns? Which would be the expected results using non-modified magnetic nanoparticles? It is a key point of your design strategy that you should highlight in this section.

Very attractive and well-performed work. Congratulations.

Author Response

Attached a file.

Round 2

Reviewer 1 Report

The authors have addressed my comments and I am happy to support it for publication.

Reviewer 2 Report

Dear authors,

thank you for the modifications you have made on your text.

Sincerely yours

Reviewer 3 Report

My comments were well addressed by the authors, so I recommend the publication of the article in this form